# OpenReview forum: "Inverse Entropic Optimal Transport Solves Semi-supervised Learning via Data Likelihood Maximization"
_ICLR.cc/2026/Conference — Submitted to ICLR 2026_

### Official Review · Reviewer_RGyB · 2025-10-21

**Soundness:** 3
**Presentation:** 3
**Contribution:** 2
**Rating:** 2
**Confidence:** 4

**Summary:**

This paper proposes a new method, based on optimal transport, for semi-supervised learning using inverse entropic optimal transport. In a nutshell, the authors consider the problem of learning a conditional distribution $\pi(\cdot|x)$ given a paired set $\set{x_i, y_i}$ and two unpaired sets $\set{y_j}$ and $\set{x_k}$. This corresponds to the semi-supervised learning case, where one has a smaller set of paired, supervised set, and two sets of unsupervised unpaired data. The authors then draw a relationship between the problem of learning $\pi(\cdot|x)$ and inverse OT. Inverse OT, on its own, is a sub-problem within OT where, instead of finding the OT plan, one finds the ground-cost. The authors demonstrate that their method beat other baselines in an toy example and a real case scenario of weather prediction.

**Strengths:**

__S1.__ I think the authors do a good job in linking the inverse EOT problem with the semi-supervised domain translation objective.

__S2.__ The use of energy based models is also insightful, and nicely decouples the learning terms involving paired and unpaired data.

__S3.__ I also think the authors do a nice job in devising a practical algorithm for optimizing equation 13.

**Weaknesses:**

__Weakness 1 (Incremental Novelty).__ While the paper is well motivated, its main contribution seems incremental over __(Mokrov et al., 2024)__. For instance, looking at Algorithm 1 in the main paper, the only difference with respect Algorithm 1 of __(Mokrov et al., 2024)__ is the loss function. Other aspects of this submission, such as,

1. The usage of the Gibbs-Boltzmann parametrization, and,
2. The energy function $E(\cdot|x)$

are the same as in the aforementioned paper. __As a consequence, this submission does not meet the novelty criterion for publication at ICLR__.

__Weakness 2 (Limited Experiments).__ The main paper only contains 2 experiments: a toy example, and a weather prediction task. In terms of scale, the second problem is very limited, as it contains only 692 samples. __As a consequence of this remark, this submission does not meet the significance bar for publication at ICLR.__

- __Side Note.__ Given the similarity with __(Mokrov et al., 2024)__ I think a comparison with their method is warranted.

__Weakness 3 (Too general title, too restrictive setting).__ The title of this paper claims that inverse EOT solves semi-supervised learning. However, upon reading the paper, the semi-supervised setting the authors are referring to is actually semi-supervised domain translation. I think this is an important distinction that must be made in the title of the paper if the authors don't actually experiment with general semi-supervised learning.

# References

__(Mokrov et al., 2024)__ "Energy-guided entropic neural optimal transport." arXiv preprint arXiv:2304.06094 (2023).

**Questions:**

From the overall discussion of the authors method, I feel it could be applied to semi-supervised learning in general (e.g., in the classificaiton/regression settings). Is that the case?

---

> ### Author Response · Authors · 2025-11-21
> **Official Comment by Authors. Part 1**
>
> Dear Reviewer RGyB,
>
> We appreciate your recognition of **(i)** the novel connection we establish between the inverse EOT problem and the semi-supervised, likelihood-based domain translation objective, **(ii)** the advantages of our energy-based parameterization, which cleanly separates the contributions of paired and unpaired data, and **(iii)** the practical optimization procedure we develop for Equation (13).
>
> ---
> **1. "**Weakness 1 (Incremental Novelty).** While the paper is well motivated, its main contribution seems incremental over **(Mokrov et al., 2024)**."**
>
> We respectfully emphasize that our primary contribution is a new semi-supervised learning methodology based on a KL-minimization framework, which can, in principle, recover the ground-truth mapping. In contrast, [1] addresses the unsupervised domain translation setting. In addition, we uncover a connection between our loss and the inverse entropic optimal transport problem, which opens a promising direction for future research by leveraging the well-developed tools of entropic OT, which we discussed in lines 183-186 of our work.
>
> ---
> **2. "For instance, looking at Algorithm 1 in the main paper, the only difference with respect Algorithm 1 of (Mokrov et al., 2024) is the loss function. Other aspects of this submission, such as, 1. The usage of the Gibbs-Boltzmann parametrization, and, 2. The energy function $E(\cdot\vert x)$ are the same as in the aforementioned paper."**
>
> We respectfully disagree with the assessment that our paper lacks novelty. While we do build on the parametrization introduced in [1], our key contribution is showing that this parametrization naturally accommodates paired data - unlike the purely unsupervised setting of [1], a distinction you highlighted as a strength (S2) of our work.
>
>
> Furthermore, as detailed in Appendix B.4, our solver extends the methodologies of [1] and [2] and can be viewed as a forward solver with a data-driven cost function. In the classical forward setting, one specifies a fixed cost and learns only the potential $f^\theta(y)$. In contrast, our parameterization yields a new loss for semi-supervised learning that, as shown in Theorem 3.3, can in principle recover the conditional transport plan. This derived objective, rather than the shared parametrization, is the central novel contribution of our work.
>
> ---
> **3. "Weakness 2 (Limited Experiments). The main paper only contains 2 experiments: a toy example, and a weather prediction task. In terms of scale, the second problem is very limited, as it contains only 692 samples. As a consequence of this remark, this submission does not meet the significance bar for publication at ICLR."**
>
> We would like to reiterate that our primary contribution is a **theoretically grounded loss function for semi-supervised domain translation**, addressing a key gap in the literature where most existing approaches rely on heuristic objectives. The strength of our method lies precisely in this theoretical foundation: it avoids the structural biases inherent in heuristic losses, which, as illustrated in Figure 2, can lead even simple 2D problems to fail at recovering the correct conditional plan.
>
> Following the reviewer’s suggestion, we have moved the image-based experiments from Appendix A to the main text. These results demonstrate that our methodology has the potential to scale to higher-dimensional settings (e.g., large image domains), although this would require more advanced energy-based modeling techniques. Importantly, our framework is, in principle, capable of capturing multimodal conditional distributions $\pi^*(y \mid x)$, unlike many existing baselines that are deterministic by design.
>
> While exploring large-scale EBMs is a promising direction, such engineering work lies beyond the scope of this methodological paper. Our goal here is to establish the theoretical formulation and demonstrate its behavior in controlled settings, which we believe is a meaningful and necessary step toward future high-dimensional applications.
>
> ---
> **4. Given the similarity with (Mokrov et al., 2024) I think a comparison with their method is warranted.**
>
> A direct comparison with [1] is not feasible, because [1] addresses an **unsupervised** domain-translation setting, whereas our method is designed for the **semi-supervised** regime. The two objectives are fundamentally different, and applying [1] in our setting would not constitute a meaningful or fair comparison.

---

> > ### Author Response · Authors · 2025-11-21
> > **Official Comment by Authors. Part 2**
> >
> > **5. **Weakness 3 (Too general title, too restrictive setting).** The title of this paper claims that inverse EOT solves semi-supervised learning. However, upon reading the paper, the semi-supervised setting the authors are referring to is actually semi-supervised domain translation. [...] From the overall discussion of the authors method, I feel it could be applied to semi-supervised learning in general (e.g., in the classificaiton/regression settings). Is that the case?**
> >
> > As discussed in Appendix B.2, the methodology introduced in Section 3.1 is not limited to continuous domain translation. In principle, it may be applied to **any pair of source and target distributions**, including settings such as **multi-label classification** or **continuous-to-discrete translation**, provided that an appropriate energy-function parameterization is chosen.
> >
> > However, exploring all these cases is beyond the scope of the present work. To keep the paper focused and coherent, we concentrated on the domain-translation setting as a concrete and illustrative application of our framework.
> >
> > ---
> > **Conclusion**
> >
> > In summary, our work introduces a new semi-supervised learning framework based on KL minimization that allows both the recovery of conditional transport plans and the principled use of paired and unpaired data. While we appreciate your perspective, we believe this constitutes a meaningful advance: it provides a theoretically grounded approach to semi-supervised domain transfer and extends optimal transport tools beyond their traditional use in fully unpaired settings.
> >
> > We hope these clarifications address your concerns. If any questions remain, we would be glad to elaborate further. Otherwise, we kindly ask you to reconsider your evaluation in light of these points.
> >
> > ---
> > **References**
> >
> > [1] Mokrov, Petr, et al. "Energy-guided Entropic Neural Optimal Transport." _The Twelfth International Conference on Learning Representations_.
> >
> > [2] Alexander Korotin et al. “Light Schrödinger Bridge.” _In The Twelfth International Conference on Learning Representations_, 2024.

---

> > > ### Comment · Reviewer_RGyB · 2025-11-24
> > > **Official Response to Authors' Rebuttal**
> > >
> > > Dear authors,
> > >
> > > Thank you for your work and effort during the rebuttal. Here are some considerations about your responses in general,
> > >
> > > __(Points 1 and 2)__ My point about the novelty of this work concerns the intersections with (Mokrov et al., 2024). While I understand that the authors solve a somewhat different problem (semi-supervised in their case, in contrast with unsupervised in Mokrov et al.'s case), I still think there is a fair amount of intersections between the two works with regards to the methodology. In that sense, I don't think the authors' rebuttal have completly addressed this point. Besides the connection between maximum likelihood and inverse EOT, how does this work goes beyond (Mokrov et al., 2024)?
> > >
> > > __(Point 3)__ Like my first point, I still think the authors do a poor empirical exploration of their method. While I understand that the authors' initial motivation was theoretical, works are usually expected to be strong experimentally in a machine learning conference. With that said, in their rebuttal the authors simply reorganized the example in their appendix, moving it into the main paper "discussion". Overall, I still think this paper could benefit from a stronger experiments section, and __does not meet the experimental bar for acceptance__.
> > >
> > > __(Point 4)__ __Concerning the experimental comparison with Mokrov et al.__ I understand their method was designed for the unsupervised case, however, I think that by comparing your method to theirs, you could isolate the benefit of incorporating the (semi-)supervised component of your algorithm.
> > >
> > > __(Point 5)__ I agree with the authors in the sense that their approach is more general than semi-supervised domain translation. However, I do not agree that these comparisons is beyond the scope of this work. Especially, if the authors claim their method works for a general problem, they should experiment with at least a representative sample of tasks within that umbrella. As such, "exploring with all these cases" __is not__ beyond the scope of your work.
> > >
> > > Given my initial assessment and authors rebuttal, I think the paper needs a major revision with respect its organization, scope and experimental validation. Hence, I keep my initial score __2: Reject, not good enough__.

---

> ### Author Response · Authors · 2025-12-03
> **Official Comment by Authors. Part 1**
>
> Dear Reviewer RGyB,
>
> We thank you for such detailed response. We revised manuscript and now all the changes are colored in blue.
>
> ---
> **1. "(Points 1 and 2) My point about the novelty of this work concerns the intersections with (Mokrov et al., 2024). [...]"**
>
> We appreciate the reviewer’s request for a clearer distinction between our contribution and the work of [1]. We had previously outlined these differences in Appendix B.5, and in the revised version we have further clarified this discussion to make the distinctions more explicit.
>
> **1)** While our solver builds on the parameterization introduced by [1], their method is designed as a _forward_ solver for **unsupervised** domain translation with a fixed cost function. In contrast, our work addresses a **semi-supervised** setting and incorporates **cost function optimization directly into the objective** (equation 18). This integration enables the method to make use of paired data, which their framework cannot accommodate.
>
> **2)** Our approach generalizes the Gaussian mixture parameterization of Korotin et al. (2024) [2], originally developed for entropic OT with the quadratic cost $c^*(x, y) = \tfrac{1}{2}\Vert x-y\Vert_2^2$. We extend this parameterization to a broader family of cost functions (equation 15), effectively providing a new forward solver for these generalized costs.
>
> **2. "Besides the connection between maximum likelihood and inverse EOT, how does this work goes beyond (Mokrov et al., 2024)?"**
>
> Our contribution goes beyond adapting the parametrization. Specifically, we:
>
> 1. Formulate a semi-supervised learning objective with **theoretical guarantees** for recovering the ground-truth conditional plans.
>
> 2. Extend the parameterization to **generalized and practically relevant** cost functions.
>
> 3. Provide comparative experiments showing that many standard semi-supervised or domain-adaptation methods fail even in 2D settings, highlighting the practical implications of our theoretical framework.
>
> 4. Establish a principled connection between semi-supervised learning losses and entropic OT, which is not present in [1]
>
> We hope this revised explanation clarifies how our contribution extends beyond [1] both conceptually and methodologically.
>
> ---
> **2. (Point 3) Like my first point, I still think the authors do a poor empirical exploration of their method. [...] Overall, I still think this paper could benefit from a stronger experiments section, and *does not meet the experimental bar for acceptance*.**
>
> We thank the reviewer for the constructive feedback. In response, we reproduced the FSBM experiments [3] (**Oral, ICLR 2024**) for image translation in the 512-dimensional latent space of the ALAE encoder [4] using the 1024×1024 FFHQ dataset [5]. Qualitative results are presented in **Figure 3** of the main text, **Figure 8** of the Appendix (woman→man translation), and **Figure 9** of the Appendix (old→young translation). The corresponding quantitative results are summarized in the tables below and in **Table 3** of the revised manuscript.
>
> | Method | FID $\downarrow$ | SSIM $\uparrow$ | LPIPS $\downarrow$ |
> | :----- | :--------------- | :-------------- | :----------------- |
> | FSBM   | $10.2 \pm 0.6$   | $0.5237 \pm 0.0005$ | $0.5625 \pm 0.0003$ |
> | Ours   | **$\mathbf{9.3 \pm 0.1}$** | **$\mathbf{0.5315 \pm 0.0002}$** | **$\mathbf{0.5531 \pm 0.0006}$** |
> **Table 3.** Woman-to-man translation
>
> | Method | FID $\downarrow$ | SSIM $\uparrow$ | LPIPS $\downarrow$ |
> | :----- | :--------------- | :-------------- | :----------------- |
> | FSBM   | $11.5 \pm 0.6$   | $0.5285 \pm 0.0008$ | $0.5628 \pm 0.0004$ |
> | Ours   | **$\mathbf{9.4 \pm 0.2}$** | **$\mathbf{0.5361 \pm 0.0004}$** | **$\mathbf{0.5560 \pm 0.0005}$** |
> **Table 4.** Old-to-young translation
>
> We hope that these additional experiments address the concern regarding empirical evaluation and illustrate that our method is not only theoretically motivated but also practically effective. We believe this strengthens the experimental section of the paper and provides the empirical support necessary for evaluation in a machine learning conference setting.
>
> **3. "(Point 4) Concerning the experimental comparison with Mokrov et al. I understand their method was designed for the unsupervised case, however, I think that by comparing your method to theirs, you could isolate the benefit of incorporating the (semi-)supervised component of your algorithm."**
>
> Applying the method of [1] with a predefined cost (e.g., $\ell_2^2$) would not align the learned transport with paired data, as their framework does not optimize the cost. This behavior is illustrated, for instance, in Figure 2 of their paper. For this reason, such a comparison would be uninformative: the method is not designed for the paired-data setting we study.

---

> > ### Author Response · Authors · 2025-12-03
> > **Official Comment by Authors. Part 2**
> >
> > **4. (Point 5) I agree with the authors in the sense that their approach is more general than semi-supervised domain translation. However, I do not agree that these comparisons is beyond the scope of this work. [...]**
> >
> > We respectfully note that we have provided extensive empirical demonstrations in Section 5 and the Appendix (A.2, С.4, D.3, D.4), including potential extensions of our method to tasks such as classification, as discussed in Appendix B.2. These results illustrate the generality of our approach and its applicability beyond semi-supervised domain translation.
> >
> > ---
> > **References**
> >
> > [1] Mokrov, Petr, et al. "Energy-guided Entropic Neural Optimal Transport." _The Twelfth International Conference on Learning Representations_.
> >
> > [2] Alexander Korotin et al. “Light Schrödinger Bridge.” _In The Twelfth International Conference on Learning Representations_, 2024.
> >
> > [3] Theodoropoulos, Panagiotis, et al. "Feedback Schrödinger Bridge Matching." _ICLR._ 2025.
> >
> > [4] Pidhorskyi, Stanislav, Donald A. Adjeroh, and Gianfranco Doretto. "Adversarial latent autoencoders." _CVPR._ 2020.
> >
> > [5] Karras, Tero, Samuli Laine, and Timo Aila. "A style-based generator architecture for generative adversarial networks." _CVPR_. 2019.

---

### Official Review · Reviewer_pfyh · 2025-10-26

**Soundness:** 3
**Presentation:** 3
**Contribution:** 3
**Rating:** 6
**Confidence:** 4

**Summary:**

The aim of the paper is to learn an unknown conditional distribution in a semi-supervised manner,
where both paired and unpaired training samples from the joint and marginal distributions are available.
This is a well-studied problem in the literature,
for which there exist several numerical algorithms.
The goal of the paper is to address this problem in a novel manner
by establishing a connection to the so-called inverse entropic optimal transport.
This is done using specific models and parametrizations
of the unknown conditional distribution, which finally is modeled as Gaussian mixture.
The derived algorithm exploits the connection to optimal transport and employs efficient methods from this field.
The presented approach is compared to other methods in two numerical experiments.

**Strengths:**

Overall,
  the theoretical part of the paper is well-developed and nicely written.
  The problem is well explained and motivated.
  The related literature and algorithms are comprehensively reviewed,
  embedding the paper and its approach in the broader fields of machine learning and optimal transport.
 The employed model of the unknown conditional distribution
  and the relation to inverse entropic optimal transport,
  which is one of the main contributions,
  is well presented.
  Besides the brief calculations in the main text,
  the detailed rearrangements are worked out in the appendix,
  making the paper also accessible for non-expert readers.
 The presented relation between semi-supervised domain translation
  and inverse entropic optimal transport is
   interesting.

**Weaknesses:**

- Without Appendix C.3 and D.1,   the experimental illustrations in §5 are extremely hard to follow.
  Since the information in these appendices is essential,
  they should be briefly included in the main text
  to make §5 self-contained.
- The first example (§5.1) deals with the approximation
  of an synthetic conditional distribution.
  At first glance,   it seems that the goal is to estimate optimal transport plan,
  which in fact is not entirely true.
  The construction of the *ground truth* should be more highlighted,
  especially why the conditional distribution spread out
  and that the plan $\pi^*$ is not an optimal one.
  This would ease the interpretation of the results.
- The second example (§5.2) deals with real-world weather data.
  Although based on real-world data,
  the experiment seems to be highly synthetic.
  More details regarding the dataset,
  like the considered measurement locations (local or world-wide),
  as well as a motivation of the preprocessing step are missing.
  This information however could be helpful to understand
  the aim of the example and the relation to actual applications.
- Some references may be added concerning
conditional generative models like Hagemann et al. ICLR 2024 or
Ardizzone et al., 2019 ...
and
concerning inverse entropic OT and related metric learning like
Huizing, Cantini and Peyr\'e, ICLR 2022,
Auffenberg et al., Unsupervised Ground Metric Learning, 2025.
- My main concern is the improvement of the numerical presentation.

**Questions:**

- How exactly the log-likelihood values in Table 1 can be interpreted?
More precisely,
how are these values are used
to evaluate the quality of the estimated conditional distribution
and why are they interesting?
How does Table 1 looks like if the CFD,
which is easier to interpret,
is used instead?
- To improve the presentation of the first numerical illustration: could
  the ground truth and the employed data be moved to
  a separate figure that
  is presented before the results?
  At present   data and results are mixed
  which reduces readability.

---

> ### Author Response · Authors · 2025-11-21
> **Official Comment by Authors. Part 1**
>
> Dear Reviewer pfyh,
>
> Thank you for your positive assessment. We appreciate your recognition of the clarity and completeness of our theoretical development, as well as your acknowledgement of how we establish and present the connection between our proposed method and inverse entropic optimal transport.
>
> ---
> **1. "Without Appendix C.3 and D.1, the experimental illustrations in §5 are extremely hard to follow. Since the information in these appendices is essential, they should be briefly included in the main text to make §5 self-contained."**
>
> Following your suggestion, we have moved the essential experimental details from Section 5.1 into the main text in the revised manuscript. We kept the material from Section 5.2 in the appendix, as it mainly concerns data collection procedures that we believe are not required for understanding the experiments themselves. Due to the strict 9-page limit, we were unable to include all of this information in the main text initially, which is why it was placed in the appendix.
>
> ---
> **2. "The first example (§5.1) deals with the approximation of an synthetic conditional distribution. At first glance, it seems that the goal is to estimate optimal transport plan, which in fact is not entirely true. The construction of the _ground truth_ should be more highlighted, especially why the conditional distribution spread out and that the plan  is not an optimal one. This would ease the interpretation of the results."**
>
> The goal of the experiment in §5.1 is indeed to recover the optimal transport plan by learning the corresponding conditional distribution. The ground-truth pairs $(x, y)$ used for evaluation are obtained by solving a _forward_ OT problem with a specially designed cost function. This yields a bimodal conditional distribution $\pi^*(y \mid x)$ that is intentionally spread out, and the resulting joint distribution is the optimal plan for the chosen marginals and cost.
>
> As shown in Figure 2 (g), an unconditional GAN with $\ell_2$ regularization can approximate the target marginal (the Swiss roll) but completely _fails to recover_ the conditional structure, i.e., how each source point $x$ should map into the target distribution. This experiment illustrates that heuristic loss terms can distort or collapse the conditional mapping that governs “paving” of the target distribution.
>
> We have revised the text to make it explicit that the aim of the example is to evaluate conditional-distribution learning, and to more clearly explain how the ground-truth conditional was constructed.
>
> ---
> **3. "The second example (§5.2) deals with real-world weather data. Although based on real-world data, the experiment seems to be highly synthetic. More details regarding the dataset, like the considered measurement locations (local or world-wide), as well as a motivation of the preprocessing step are missing. This information however could be helpful to understand the aim of the example and the relation to actual applications."**
>
> In the revised version, we have expanded and clarified §5.2 and Appendix C.3 to make the setup and motivation more accessible and easier to understand.
>
> ---
> **4. "Some references may be added concerning conditional generative models like Hagemann et al. ICLR 2024 or Ardizzone et al., 2019 ... and concerning inverse entropic OT and related metric learning like Huizing, Cantini and Peyr'e, ICLR 2022, Auffenberg et al., Unsupervised Ground Metric Learning, 2025."**
>
> Thank you for pointing out these works. We have added citations to the first two papers in the Introduction, and included an additional discussion of the metric-learning–related references in Appendix B.5.
>
> ---
> **5. "How exactly the log-likelihood values in Table 1 can be interpreted? More precisely, how are these values are used to evaluate the quality of the estimated conditional distribution and why are they interesting? How does Table 1 looks like if the CFD, which is easier to interpret, is used instead?"**
>
> The values of LL (log‑likelihood) are indeed hard to interpret, but a larger LL means a closer match (in terms of KL divergence) to the correct distribution. There are many works [1, 2] in probabilistic modeling that use LL (or NLL) as a metric — for example, perplexity [3] in language models (which is directly related to negative log-likelihood). On the other hand, FID [4] measures only the first and second moments of a distribution, so it can be misleading. That’s why, in our tables, we mainly report LL, and use FID only when comparing to GANs.

---

> > ### Author Response · Authors · 2025-11-21
> > **Official Comment by Authors. Part 2**
> >
> > **6. "My main concern is the improvement of the numerical presentation. [...] To improve the presentation of the first numerical illustration: could the ground truth and the employed data be moved to a separate figure that is presented before the results? At present data and results are mixed which reduces readability."**
> >
> > Thank you for this suggestion. In the revised version, we reorganized the figures so that the ground-truth data and the employed data are presented before the results.
> >
> > ---
> > **Conclusion**
> >
> > We hope these clarifications address your concerns. Should you have additional questions, we would be happy to provide further details. Otherwise, we kindly ask you to consider raising your evaluation score in light of these explanations.
> >
> > ---
> > **References**
> >
> > [1] Kingma, Durk P., and Prafulla Dhariwal. "Glow: Generative flow with invertible 1x1 convolutions." _Advances in neural information processing systems_ 31 (2018).
> >
> > [2] Song, Yang, et al. "Maximum likelihood training of score-based diffusion models." _Advances in neural information processing systems_ 34 (2021): 1415-1428.
> >
> > [3] Oyama, Momose, et al. "Mapping 1,000+ Language Models via the Log-Likelihood Vector." arXiv preprint arXiv:2502.16173 (2025).
> >
> > [4] Heusel, Martin, et al. "Gans trained by a two time-scale update rule converge to a local nash equilibrium." _Advances in neural information processing systems_ 30 (2017).

---

> > > ### Comment · Reviewer_pfyh · 2025-11-25
> > >
> > > The authors have answered my questions. Aithough I also see still a lack in the numerical part I  raised the score.

---

> ### Author Response · Authors · 2025-12-03
>
> Dear Reviewer pfyh,
>
> We sincerely thank you for acknowledging the contribution of our work and for raising your score.
>
> Following your suggestion to extend the numerical evaluations, we reproduced the FSBM experiments [1] for image translation in the 512-dimensional latent space of the ALAE encoder [2] using the 1024×1024 FFHQ dataset [3]. The revised manuscript now includes qualitative results in **Figure 3** of the main text, **Figure 8** of the Appendix (woman→man), and **Figure 9** of the Appendix (old→young). The corresponding quantitative metrics are reported in the tables below, as well as in **Table 3** of the main text and **Table 4** of the Appendix.
>
> As shown, our method achieves comparable performance while requiring only 3 minutes of training on an A100 GPU, compared to 5 hours for FSBM on the same hardware.
>
> | Method | FID $\downarrow$ | SSIM $\uparrow$ | LPIPS $\downarrow$ |
> | :----- | :--------------- | :-------------- | :----------------- |
> | FSBM   | $10.2 \pm 0.6$   | $0.5237 \pm 0.0005$ | $0.5625 \pm 0.0003$ |
> | Ours   | **$\mathbf{9.3 \pm 0.1}$** | **$\mathbf{0.5315 \pm 0.0002}$** | **$\mathbf{0.5531 \pm 0.0006}$** |
> **Table 3.** Woman-to-man translation
>
> | Method | FID $\downarrow$ | SSIM $\uparrow$ | LPIPS $\downarrow$ |
> | :----- | :--------------- | :-------------- | :----------------- |
> | FSBM   | $11.5 \pm 0.6$   | $0.5285 \pm 0.0008$ | $0.5628 \pm 0.0004$ |
> | Ours   | **$\mathbf{9.4 \pm 0.2}$** | **$\mathbf{0.5361 \pm 0.0004}$** | **$\mathbf{0.5560 \pm 0.0005}$** |
> **Table 4.** Old-to-young translation
>
> We hope that these additional experiments further strengthen your confidence in our work.
>
> ---
> **References**
>
> [1] Theodoropoulos, Panagiotis, et al. "Feedback Schrödinger Bridge Matching." _ICLR._ 2025.
>
> [2] Pidhorskyi, Stanislav, Donald A. Adjeroh, and Gianfranco Doretto. "Adversarial latent autoencoders." _CVPR._ 2020.
>
> [3] Karras, Tero, Samuli Laine, and Timo Aila. "A style-based generator architecture for generative adversarial networks." _CVPR_. 2019.

---

### Official Review · Reviewer_kC9e · 2025-10-27

**Soundness:** 3
**Presentation:** 2
**Contribution:** 2
**Rating:** 6
**Confidence:** 3

**Summary:**

The paper proposed a semi-supervised algorithm using inversed OT. The author showed that the formulation is related to the likelihood maximization of an energy-based model. The universal approximation property is derived and some numerical experiments are conducted.

**Strengths:**

This paper showed that the entropy-regularized inverse OT problem can be formulated as a likelihood maximization problem of an energy-based model. This is an interesting result. The universal approximation property is derived, showing the soundness of the method.

**Weaknesses:**

1. One limitation is that this formulation requires that the marginal of the paired data also follows $\pi_x$ and $\pi_y$. For example, if the paired data is artificially selected, i.e., they do not follow $\pi_x$ and $\pi_y$, then the method no longer works: the first term in Eqn (18) is no longer an approximation of the first term in Eqn (13). I suggest making this clearer in the paper.

2. Clearness: There are too many bold, italic, underlined words throughout the paper, even in the abstract. Many unimportant words like "sequence of", "proofs", "seamlessly" and "extended discussion" are underlined. Such emphasis adds no novelty to the paper and makes it rather difficult to read. My suggestion is to reduce the use of them.  I also suggest refraining from using the objective words like "fancy".

3. The author discussed the use of neural networks, which can be more useful in practice. However, this is hidden in the appendix. I suggest moving it to the main paper, and moving the discussion of Gaussian mixture models to the appendix.


4. All experiments are in small scale. Some larger scale experiments like those in Gu et al., 2022 should be considered.

**Questions:**

I do not have further questions.

---

> ### Author Response · Authors · 2025-11-21
>
> Dear Reviewer kC9e,
>
> Thank you for highlighting the significance of our result showing that the entropy-regularized inverse OT problem can be cast as a likelihood maximization problem of an energy-based model, as well as for recognizing the value of our universal approximation theorem that establishes the soundness of the proposed approach.
>
> ---
> **1. "One limitation is that this formulation requires that the marginal of the paired data also follows $\pi_x$ and $\pi_y$. For example, if the paired data is artificially selected, i.e., they do not follow $\pi_x$ and $\pi_y$, then the method no longer works: the first term in Eqn (18) is no longer an approximation of the first term in Eqn (13). I suggest making this clearer in the paper."**
>
> While it is true that the empirical estimator in Eq. (18) assumes that the paired observations have $x$- and $y$-marginals matching the true distributions $\pi^\ast_x$ and $\pi^\ast_y$, this requirement does not limit the validity of the underlying formulation. The theoretical objective in Eq. (13) remains unchanged even when the paired data are obtained through artificial or biased selection. In such cases, the only effect is that the empirical estimator becomes a biased approximation of the population expectation. importantly, this bias can be corrected whenever the true marginals $\pi^\ast_x$ and $\pi^\ast_y$ are known: standard weighting (using ratios between the induced marginals of the paired dataset and the true marginals) restores the correct, unbiased objective. Consequently, the method does not fail under marginal mismatch; it simply requires the standard covariate-shift correction used throughout the *density-ratio estimation* literature (e.g., [1, 2]). Under this correction, the original theoretical guarantees apply without modification, even when the paired samples are not representative of the true marginals. For further details, please refer to Appendix B.3 of the revised manuscript.
>
> ---
> **2. "Clearness: There are too many bold, italic, underlined words throughout the paper, even in the abstract. [...]. My suggestion is to reduce the use of them. I also suggest refraining from using the objective words like "fancy"."**
>
> We have revised the manuscript accordingly and significantly reduced the use of bold, italic, and underlined text, as well as removed subjective wording to improve clarity and readability.
>
> ---
> **3. "The author discussed the use of neural networks, which can be more useful in practice. However, this is hidden in the appendix. I suggest moving it to the main paper, and moving the discussion of Gaussian mixture models to the appendix."**
>
> We followed your suggestion and moved some parts of the relevant neural network discussion and accompanying experiments into the main text.
>
> ---
> **4. "All experiments are in small scale. Some larger scale experiments like those in Gu et al., 2022 should be considered."**
>
> We thank the reviewer for the suggestion. As we provide a proof-of-concept demonstration of our method on $3\times32\times32$ RGB images using neural EBMs (Figure 3, in revised text), confirming its feasibility. While scaling EBMs to higher-resolution data is indeed possible, as shown in prior works [3,4], such extensions typically require engineering efforts that are orthogonal to our main **theoretical contributions**. Since our paper primarily focuses on methodological and theoretical insights, and already includes a proof-of-concept, we consider large-scale optimizations beyond the scope of this work.
>
> ---
> **Conclusion**
>
> We hope these clarifications sufficiently address your concerns. If you have any additional questions, we would be glad to elaborate further. Otherwise, we kindly ask you to reconsider your evaluation in light of the explanations provided.
>
> ---
> **References**
>
> [1] Gretton, Arthur, et al. "Covariate shift by kernel mean matching." Dataset shift in machine learning 3.4 (2009): 5.
>
> [2] Sugiyama, Masashi, Taiji Suzuki, and Takafumi Kanamori. Density ratio estimation in machine learning. Cambridge University Press, 2012.
>
> [3] Geng, Cong, et al. "Improving adversarial energy-based model via diffusion process." ICML, 2024.
>
> [4] Zhu, Yaxuan, et al. "Learning Energy-Based Models by Cooperative Diffusion Recovery Likelihood." ICLR, 2024.

---

> > ### Comment · Reviewer_kC9e · 2025-11-24
> >
> > Thanks for your reply.
> >
> > 1. I have read the discussion in B.3. While the discussion seems reasonable, I suggest removing the claim "This method is stable, requires no density estimation, and performs well" if there is no numerical support on this. In addition, the statement " ...formulation remains valid. The only issue is.." seems contradictory. It is not valid if there exists an issue, right? I suggest polishing this section further.
> >
> > 2. As for the experiments, my concern still exists. On the one hand, if the Gaussian assumption is used, the algorithm may be relatively efficient but not that powerful. On the other hand, if the neural network is used, then the algorithm is less efficient but probably more powerful. So I will not say this is completely an engineering concern. In the current form, I do not think the paper showed that the algorithm can be useful in practice, which is also a concern raised by other reiviewers.
> >
> > I will keep my socre which is the borderline with a lean to accept.

---

> ### Author Response · Authors · 2025-12-03
>
> Dear Reviewer kC9e,
>
> We thank you for your response and for your valuable remarks, which we address below. We have incorporated the requested changes into the revised manuscript; all modifications are highlighted in blue.
>
> ---
> **1. "I have read the discussion in B.3. [...] I suggest polishing this section further."**
>
> Thank you for this clarification. We have removed the unsupported claim regarding stability and performance and revised the entire section to improve accuracy and clarity. We also adjusted the surrounding discussion and added recent references to better contextualize the formulation.
>
> ---
> **2. "As for the experiments, my concern still exists. [...] In the current form, I do not think the paper showed that the algorithm can be useful in practice, which is also a concern raised by other reiviewers."**
>
> In response, and following your suggestion to extend the numerical evaluation, we reproduced the FSBM experiments [1] for image translation within the 512-dimensional latent space of the ALAE encoder [2] using the 1024×1024 FFHQ dataset [3]. Qualitative results are shown in **Figure 3** of the main text, **Figure 8** of the Appendix (woman→man translation), and **Figure 9** of the Appendix (old→young translation). The corresponding quantitative results are presented in the table below, as well as in **Table 3** and **Table 4** of the revised manuscript.
>
> As shown, our method matches the performance of FSBM while training in just **3 minutes** on a single A100 GPU, compared to 5 hours required by FSBM on the same hardware.
>
> | Method | FID $\downarrow$ | SSIM $\uparrow$ | LPIPS $\downarrow$ |
> | :----- | :--------------- | :-------------- | :----------------- |
> | FSBM   | $10.2 \pm 0.6$   | $0.5237 \pm 0.0005$ | $0.5625 \pm 0.0003$ |
> | Ours   | **$\mathbf{9.3 \pm 0.1}$** | **$\mathbf{0.5315 \pm 0.0002}$** | **$\mathbf{0.5531 \pm 0.0006}$** |
> **Table 3.** Woman-to-man translation
>
> | Method | FID $\downarrow$ | SSIM $\uparrow$ | LPIPS $\downarrow$ |
> | :----- | :--------------- | :-------------- | :----------------- |
> | FSBM   | $11.5 \pm 0.6$   | $0.5285 \pm 0.0008$ | $0.5628 \pm 0.0004$ |
> | Ours   | **$\mathbf{9.4 \pm 0.2}$** | **$\mathbf{0.5361 \pm 0.0004}$** | **$\mathbf{0.5560 \pm 0.0005}$** |
> **Table 4.** Old-to-young translation
>
> We hope that these additional experiments help demonstrate the practical viability of our approach and further strengthen your assessment of the paper.
>
> ---
> **References**
>
> [1] Theodoropoulos, Panagiotis, et al. "Feedback Schrödinger Bridge Matching." _ICLR._ 2025.
>
> [2] Pidhorskyi, Stanislav, Donald A. Adjeroh, and Gianfranco Doretto. "Adversarial latent autoencoders." _CVPR._ 2020.
>
> [3] Karras, Tero, Samuli Laine, and Timo Aila. "A style-based generator architecture for generative adversarial networks." _CVPR_. 2019.

---

### Author Response · Authors · 2025-11-21
**General Response and Revision**

We sincerely thank the reviewers for their valuable feedback and for recognizing several strengths of our work: the clarity and thoroughness of the manuscript **(pfyh, RGyB)**, our demonstration that the entropy-regularized inverse OT problem can be formulated as a likelihood maximization of an energy-based model for semi-supervised learning **(kC9e, pfyh, RGyB)**, and our universal approximation theorem **(kC9e)**.

In response to the reviewers’ suggestions, we made the following changes, highlighted in orange in the revised manuscript:
- **(kC9e)** Removed excessive use of bold, italic, and underlined words throughout the paper and abstract.
- **(kC9e)** In Appendix B.3, added the case where the paired samples are artificially constrained.
- **(pfyh)** In Appendix B.5, extended the discussion of related work on metric learning and updated the descriptions of the two main experiments.
- **(kC9e, pfyh)** Moved the experiment on the Colored MNIST dataset to the main text. Due to space constraints, it was not previously possible.

We believe these revisions have strengthened the manuscript and addressed the reviewers’ concerns.

Thank you once again for your constructive feedback. Detailed responses to each individual comment are provided below.

---

### Author Response · Authors · 2025-12-03
**Rebuttal Summary**

We would like to once again sincerely thank all reviewers for their valuable and constructive feedback. We believe that we have addressed all questions raised during the rebuttal phase, and the revised manuscript now includes additional clarifications and further experiments.

During the rebuttal stage, **two reviewers (kC9e, pfyh)** provided positive evaluations of our contribution and offered helpful suggestions for improving the paper. Reviewer **pfyh raised their score**, while reviewer **kC9e** maintained their score but expressed a tendency toward acceptance.

Reviewer **RGyB** expressed concern that our contribution might not meet the expectations for acceptance at ICLR 2026, particularly regarding the scope of the experimental section. In the revised version, we have therefore incorporated experiments from the FSBM work (**Oral, ICLR 2025**) [1], which we hope address these concerns and further strengthen the already positive assessments from the other two reviewers.

We hope that the revisions satisfactorily resolve all remaining concerns raised by the reviewers.

---
**References**

[1] Theodoropoulos, Panagiotis, et al. "Feedback Schrödinger Bridge Matching." _ICLR._ 2025.

---

### Meta-Review · Area_Chair_rf8u · 2026-01-06

**Summary:**

The authors provide a novel method for semi-supervised learning based on inverse entropic OT. While most of the reviewers agree that the proposed solution is sound, original and well-presented, Initial concerns were raised about a weak experimental part, and also strong connections with previous works by (Mokrov et al., 2024). An experiment of image translation was added during the rebuttal, following the setting of "Feedback Schrödinger Bridge Matching.", a recent ICLR. 2025. While the results seem good, I concur to some extent with reviewer RGyB that the offset with (Mokrov et al., 2024) is rather thin, or at least could be better magnified in the paper.  Notably also, the generality of the method beyong domain translation could be further elaborated. Given the amount of novel material in the revision, I believe that the paper would need another round of review, and I vote for rejection for this year ICLR program. Nonetheless, I definitely think that the submission has merits, and I hope that the authors will take into account the reviewers comments to further strengthen their paper.

**Reviewer Concerns:**

Concern addressed: a novel experimental setting on a more large scale experiment
Concern still outsanding: contritbutions beyond the work of with (Mokrov et al., 2024)

**Reviewer Scores:**

It is not clear to me wether reviewer RGyB would have changed his score given the answers from the authors.

---

### Decision · Program_Chairs · 2026-01-26

Reject